# Transradial Embolization, an Underused Type of Uterine Artery Embolization Approach: A Systematic Review

**DOI:** 10.3390/medicina57020083

**Published:** 2021-01-20

**Authors:** Loredana Maria Himiniuc, Mara Murarasu, Bogdan Toma, Razvan Popovici, Ana-Maria Grigore, Ioana-Sadiye Scripcariu, Mihaela Oancea, Mihaela Grigore

**Affiliations:** 1Department of Obstetrics and Gynecology, University of Medicine and Pharmacy “Gr. T. Popa”, 700015 Iasi, Romania; loredanahiminiuc@gmail.com (L.M.H.); razpopovici@yahoo.com (R.P.); ana_grigore12@yahoo.com (A.-M.G.); isscripcariu@gmail.com (I.-S.S.); mihaela.grigore@edr.ro (M.G.); 2Obstetrics and Gynecology Clinical Hospital “Cuza Voda”, 700038 Iasi, Romania; mara.murarasu@gmail.com; 3Department of Obstetrics and Gynecology, University of Medicine and Pharmacy “Iuliu Hateganu”, 400012 Cluj-Napoca, Romania

**Keywords:** uterine artery embolization, transradial approach, transradial embolization, radial approach, myoma

## Abstract

*Background and Objectives:* The most utilized approach for the embolization of uterine arteries is the transfemoral path. However, the transradial approach (TRA) has been gaining popularity among cardiologic interventions in the last years but only few studies have shown its applicability in uterine myoma treatment. The objective of this paper is to assess the feasibility, safety and efficacy of TRA when compared with the transbrachial, transulnar or transfemoral approach (TFA) for uterine arteries embolization (UAE). *Materials and methods:* A systematic review of the literature that analyzes the TRA for UAE it was carried out, in order to assess its safety and effectiveness. It was systematically searched the literature (Google Scholar, PubMed/MEDLINE, Cochrane Library and Embase) using the words “uterine artery embolization”/“uterine embolization” and “transradial”/“radial”. All the relevant papers published until March 2020 were retrieved and analyzed. *Results:* Ten studies were considered eligible for this topic. TRA is a comparable method with TFA for uterine artery embolization. *Conclusions:* These studies allowed us to conclude that TRA is as safe and efficient as TFA. Its advantages include few complications, shorter hospitalization period, and rapid mobilization but a steeper learning curve has the disadvantage of a longer learning curve compared to TFA. Yet, these findings are built on few reports and more research is needed.

## 1. Introduction

Uterine artery embolization (UAE) represents a feasible therapeutic option introduced in clinical practice for females with symptomatic uterine leiomyoma who want to avoid the surgical intervention [1,2]. One of the first attempts of uterine artery embolization, as described in the literature, was performed in 1976 on two patients who were suffering from intractable metrorrhagia and were poor candidates for surgery [3], but two decades later, in 1995, Ravina et al. reported successful outcomes of UAE in 16 cases of uterine myoma [4]. Since then, several large studies have proved both the effectiveness and the safety of UAE [5,6] and current guidelines on this topic, established the impact of UAE for fertile patients who desire a pregnancy, the clinical failure risk or re-intervention risk of this procedure, but also the success of the intervention based on clinical symptomatology. Hence, women with symptomatic fibroids, with contraindications for surgery due to an associate medical condition, or who does not accept blood transfusion or who had a previous failed surgery for multiple fibroids are fine candidates for UAE procedure [7]. In addition, although uterine myoma is still the main indication for UAE, other conditions such as obstetrical hemorrhage [8], adenomyosis [9], and cervical ectopic pregnancy [10] can benefit from it.

The classic and the most utilized approach for UAE is by transfemoral catheterization. The advantages of this access site are its easy location and approach for puncture and hemostasis. When TFA is difficult, the access through other vessels, including the brachial, radial, and ulnar arteries, is especially used in cardiologic interventions. The transradial approach (TRA) has been gaining popularity over the last three decades, and nowadays represents a valuable technique for cardiovascular interventions compared to TFA, due to its lower risk of complications such as hemorrhage and vascular complications (3.7% TFA vs. 1.4% TRA, *p* < 0.0001) [11,12]. Several studies proved the same efficacy as for the TFA in peripheral arterial procedures [13], showing great benefits for patient contentment, rapid hospital discharge due to short recovery time [14]. Nevertheless, the use of the radial artery access for non-coronary interventions, especially for UAE has received less attention and very few studies have been published on this topic. This paper represents a systematic review of the literature data in order to describe the use of TRA for UAE.

## 2. Materials and Methods

It was systematically searched for the relevant literature on PubMed/MEDLINE, Cochrane library, Google Scholar, Embase and grey literature (academic papers, including theses and dissertations, research and committee reports, government reports, conference papers), using the terms “uterine artery embolization”/“uterine embolization” and “transradial”/“radial” for all research papers that were published until March 2020. Two independent reviewers extracted data and assessed the quality of the articles. It was used the Preferred Reporting Items for Systematic Reviews and Meta-Analyses (PRISMA) Statement, available by the Enhancing the Quality and Transparency of Health Research (EQUATOR) network, for all the analyses, study design, drafting, data interpretation and revision that were made.

The primary goal of this systematic review was to appreciate the procedure’s effectiveness and safety. It was considered successful when the entire embolization procedure was performed as intended, with no need to change the access route. Moreover, it was evaluated the complications that followed the local access of the catheter and their incidence rates.

## 3. Results

The literature search retrieved ten studies (eight retrospective and two prospective studies) [15,16,17,18,19,20,21,22,23,24], as further discussed below (Table 1). A conference proceeding abstract was also included between the studies [19].

In all the studies except one, the Allen test or Barbeau test was performed before the procedures in order to verify the ulnar artery’s ability to counterbalance for transitory occlusion of the radial artery. Subjects with abnormal Allen or Barbeau test results were excluded from the study. Pre-procedural sonography assessment was reported in seven studies to measure the radial artery caliber, anatomical varieties, and for guided puncture.

The 4-French or 5-French diameter catheter with 100–150 cm length was mostly used with the addition of various micro-catheters in some cases where anatomy of the vessels was found abnormal and for difficult procedures. The hemostasis was achieved using various compression devices (Table 2).

The success rate of UAE using TRA was very good among the selected studies, ranging from 95% to 100%, with an average of 98.86% as it is seen in Table 3. This is similar to TRA for cardiologic interventions, where the failure of the technique is limited (5% to 7% of cases) and is usually associated with elderly patients, women, and patients with a decreased body mass index. This increased success rate of the procedures using TRA might be because patients with uterine myoma are usually younger with no comorbidities. There was no permanent occlusion of the radial vessel reported among patients; instead, some cases of transitory radial occlusion were reported (Table 1).

## 4. Discussion

Campeau firstly described transradial access for angiographic diagnosis in 1989 in a series of 100 patients [25]. He concluded that TRA is as effective and safe as the transbrachial approach, which was the alternative to the TFA at that time. The embolization method has been used in various hemorrhagic conditions such as rectus sheath hematoma or postpartum hemorrhage [26]. Nowadays, in many centers, TRA is the main access for both coronary angiography procedures and patients who go for cardiac interventions.

Several advantages have been attributed to TRA technique. The radial artery, compared to femoral artery or even ulnar artery, is shallow and easy to compress and bleeding is rapidly controlled. Because there are no major vessels or nerves on the radial artery topography, the risk for vascular and peripheral nerve injury is reduced and complications like compartment syndrome of the arm or arteriovenous fistulas have a low incidence [27]. The estimated incidence of vascular complications varies from 1.5% to 30.5% among transradial coronary procedures [28].

The RIVAL trial, the largest randomized multicenter comparison between TRA and TFA among acute coronary syndrome patients, demonstrated that while these two approaches had similar overall safety and efficacy, the TRA for coronary angiography and percutaneous coronary interventions significantly reduced major vascular complications by decreasing the incidence of large hematoma and pseudoaneurysms requiring closure [29].

A Cochrane systematic review that compared transradial access with transfemoral access for both diagnostic and interventional methods in patients with coronary artery disease reported a decreased cardiac death rate and short-term NACE, and similarities regarding the rate of acute myocardial infarction (RR 0.91, 95% CI 0.81 to 1.02; 19,430 cases; 11 randomized controlled trials) [30].

Moreover, TRA may have a protective role in acute kidney injury development after coronary interventions. The incidence of chronic kidney disease is lower in the next 6 months after TRA procedures compared to TFA in percutaneous coronary interventions [31]. Kooiman et al. reported a significantly decreased risk of developing acute kidney injury after TRA when compared to TFA (odds ratio 0.74, 95% confidence interval 0.58 to 0.96) [32].

In a randomized clinical trial where TRA was compared to TFA among cardiac patients, Cooper et al. reported the following benefits of TRA procedure: the patient’s choice, an increased quality of life metrics, no requirement for immobilization or bladder catheterization, and reduced hospitalization costs [33].

Despite its various advantages, TRA is not used worldwide. Some highly experienced TFA operators are reluctant to retrain in TRA due to several challenges in transradial access. They include anatomical variations, small lumen, radial artery spasm, and radial artery occlusion.

Although it is known that TRA has an increased learning curve compared to TFA [34], no statistical significance was observed between the procedures’ fluoroscopic time [22], showing that like other various types of procedures that are based on the operator’s skills, TRA-operating skills refine as the experience of the operator increase [35].

Some authors suggested that patients who are shorter than 165 cm and 170 cm present an anatomically reduced size of the arterial lumen, making the access technique more exhausting and complex [18,36]. However, the present equipment increases the benefits of TRA procedure, making it suitable and successful for a heterogeneous population.

Gender-based differences in the diameter of the distal radial artery are well known. A higher rate of vascular complications and technical failure may appear in women due to the reduced diameter of the artery [37]. Although the radial artery presents the capacity to expand, its diameter is considerably reduced compared to the femoral and brachial vessels, having a lumen diameter mean smaller than 3 mm [38]. Still, Pham et al. reported a success rate of UAE via TRA of 100% in 60 women with radial artery diameter between 2 and 3 mm, reporting only one case of arterial vasospasm [19].

The smaller the radial artery caliber is, the higher the chances of artery spasm and local discomfort during the intervention are. A reduced radial artery diameter represents a risk factor for the primary patency rate of arteriovenous fistulas. Yan et al. showed that in 65 patients who underwent transradial coronary interventions, TRA procedure decreases the response to flow-mediated dilatation and nitroglycerin-mediated dilatation, determining a rapid and persistent damage of the vasodilatation function [39].

Patients with palmar arch instability present a major incidence for vessel occlusion, and TRA is not indicated in this group category. For this reason, the subjective Allen’s test and objective tests like Barbeau test or Princeps Pollicis artery ultrasound are recommended for every patient before the procedure in order to evaluate the hand collateral patency [40].

The incidences of local vascular per procedural complications in UAE by TRA were low among the selected studies, ranging from 0 to 3.3%. The permanent occlusion of the radial vessel represents the most serious complication of TRA, with an incidence that varies in the literature from 1% to 10% [41], and this is probably because its occurrence is often clinically silent due to the dual blood supply of the hand. Moreover, the slender catheters and sheaths (4-French or 5-French) used for cardiology and gynecology interventions may contribute to the low incidence of radial artery permanent occlusion. In all ten studies, no such complication was described. Hydrophilic material was especially chosen for the radial approach in order to reduce the patient’s discomfort. The technique’s failure is mainly due to the difficult artery puncturing and secondly, to the anatomical vessel variations or arterial spasm. The administration of heparin, nitroglycerin and verapamil directly through the access sheath reduces the risk of arterial vasospasm and occlusion and this may be a fine reason for the higher success rate of the procedures.

Pain is an important issue to deal with during and after the procedure. Different associations of painkillers to manage pain were described in the selected studies. One of them reported a 14% readmission rate because of the pain (Table 1).

In Thakor’s study, 98% of individuals who previous opted for a femoral access intervention would desire the transradial route for following procedures. The main reasons for this preference are as follows: rehabilitation time after discharge, earlier ambulation after the intervention, seated or Semi-Fowler’s position sufficient for the post-interventional recovery, rapid hospital discharge, and embarrassment correlated to groin preparation [21]. It is also important to note that all patients who listed modesty as a reason for TRA were women, and this should be considered especially for the gynecological field.

In cases of non-coronary interventions, very few studies regarding TRA have been published. For UAE, ten articles were retrieved that describe this access site. In all studies, the indication for UAE was uterine myoma, except one in which adenomyosis was associated too [22].

Based on this literature review, both increased success rate and decreased incidence of local complications lead us to consider the transradial route as a valid option for UAE. Even if some professionals are reluctant to use this approach, mainly due to the longer learning curve, there are some clinical situations where an alternative route to classic TFA is mandatory. These relative contraindications represent cases such as important obesity, vascular complications during previous catheterizations via the femoral approach, known peripheral vascular conditions in the inferior limbs or difficulty in remaining supine for a long period [42]. In all these cases, the transradial access could be a valid way for performing the UAE, compared with radiofrequency myolysis or others [43].

The limitations of the analyzed studies are that they are (with two exceptions) retrospective studies and non-randomized. In the future, there is a need for prospective randomized studies performed by trained radiologists to evaluate the traditional TFA versus TRA route for UAE.

## 5. Conclusions

Uterine fibroid embolization using the TRA is as safe and as effective as the TFA. Its advantages include rapid mobilization and shorter hospital stay. However, it has a longer learning curve. This conclusion relies on a few studies and subsequent reports are required.

## Figures and Tables

**Table 1 medicina-57-00083-t001:** Summary of studies that were included in the analysis.

Author, Year	Type of Study	Number of Cases and Indications	Complications
Yamashita, 2007	Retrospective	7 UAE among 380 TRA non-coronary procedures	Radial vasospasm (4 cases—1.1% of the total) Radial artery injury (1 case) 1.8% total complication rate
Resnick, 2014	Retrospective	29 UAE via TRA for symptomatic uterine fibroids	No major or minor complications No necessary to switch to TFA
Salmeron, 2015	Prospective	62 of 64 patients—bilateral UAE for uterine myoma	Unilateral UAE (2 cases of 64) No local vascular complications Early readmission for pain 14% Three patients (4.69%) went for hysterectomy after UAE
Posham, 2016	Retrospective	116 UAE for uterine myoma among 1512 TRA non-coronary procedures	Pseudo-aneurysm (1 case) Seizure (1 case) Hematoma/bleeding (13 cases) Radial artery occlusion (11 cases) Arm pain (6 cases) Radial artery spasm (6 cases) (for the whole series) 27 cases (1.8%) required TFA
Thakor, 2016	Retrospective	88 UAE for uterine myoma among 749 TRA non-coronary procedures	No complications in the cases of UAE Radial artery occlusion 0.3% Small hematoma (1 case)
Biederman, 2016	Retrospective	4 UAE for uterine myoma among 22 TRA non-coronary procedures (18.2%) in morbidly obese patients	No minor or major complications in all 22 cases
Pham, 2016	Retrospective	60 cases UAE for uterine myoma radial artery diameter 2–3 mm (mean 2.4 mm)	Ecchymosis (1 case) Radial artery spasm (1 case)
Mortensen, 2018	Prospective	39 TFA UAE vs. 27 TRA UAE for symptomatic fibroids +/− adenomyosis	No necessary switch to TFA No other complications
Gjoreski, 2019	Retrospective	11 UAE cases via TRA vs. 13 cases via TFA	Non-flow-limiting dissection of left internal iliac artery in the TFA group (1 case) Prolonged pain in the left forearm in the TRA group (1 case)
Nakhaei, 2019	Retrospective	90 (91) UAE cases via TRA vs. 92 cases via TFA—1 case switched to TFA due to vasospasm (1%)	Groin hematoma (5 cases) and groin pain (2 cases) in the TFA group Transitory focal occlusion (4 cases) and focal pain (1 case) in the TRA group

UAE—Uterine artery embolization; TRA—Transradial approach; TFA—Transfemoral approach.

**Table 2 medicina-57-00083-t002:** Materials and methods used for TRA technique.

Author, Year	Pre-Procedural Assessment	Access	Catheter or Equipment	Hemostasis
Yamashita, 2007	Allen test Radial pulse palpation	22-gauge needle, 17 cm length 4F arterial introducer sheath (Slit Super-Sheath; Medikit, Tokyo, Japan), Seldinger technique	130–150 cm length 4F KI-6 catheter (Medikit) was used for selective angiography and interventions	Manual compression for 5 min An original designed compression tourniquet comprising an elastic bandage around a small gauze roll placed on the puncture site, and fixed by elastic tape (5 h)
Resnick, 2014	Barbeau test Radial and ulnar artery sonography	21-gauge, 2.5 cm echogenic-tip needle, Ultrasound guidance, 0.021 inch tapered hydrophilic 4-F Glidesheath (Terumo, Somerset, NJ, USA)	120-cm, 4F angled tip hydrophilic-coated Glidecath (Terumo), For difficult anatomy: Renegade Hi-Flo (Boston Scientific, Natick, Massachusetts, USA) or Progreat (Terumo) micro-catheters	TR Band (Terumo Interventional Systems)
Salmeron, 2015	-	7-cm length 5F radial arterial introducer sheath	125 cm length 5F curve catheter extra-wide multipurpose, 150 cm 2.7F micro-catheter (Renegade, Bolton Scientific, EE. UU.)	TR Band (Terumo Interventional Systems)
Posham, 2016	Barbeau test	21-gauge echogenic-tip needle, Ultrasound guidance, 10-cm length Glidesheath (Terumo Interventional Systems, Somerset, NJ, USA), Seldinger technique	100–150 cm length 5F catheter, 130–150 cm standard length micro-catheters	TR Band (Terumo Interventional Systems)
Thakor, 2016	Barbeau test	21-gauge needle, Ultrasound guidance, Hydrophilic sheath (Glidesheath Slender, Terumo Medical Corporation, Somerset, NJ, USA; Prelude Ease, Merit Medical Systems, South Jordan, UT, USA), Seldinger technique	150 cm length 4F catheter, Shorter base micro-catheter up to 125 cm, Short non-flushable hemostatic valve (i.e., FLO30, Merit Medical Systems)	Compression device (i.e., Safeguard Radial, Merit Medical Systems) -TR Band (Terumo Interventional Systems) -R-Band (Vascular Solutions, Minneapolis, MN, USA)
Biederman, 2016	Barbeau test	21-gauge, 1.5″ (38 mm) echogenic tip needle (Teurmo, Somerset, NJ, USA), Ultrasound-guided puncture, Hydrophilic coated Glidesheath (Terumo)	0.021 nitinol guide wire (Terumo)	TR Band (Terumo Interventional Systems)
Pham, 2016	Barbeau test Radial artery diameter of 2.0–3.0 mm assessed by ultrasonography	Ultrasound guidance, 4F Glidesheath (Terumo Medical Corporation, Somerset, NJ, USA)	-	-
Mortensen, 2018	Barbeau test Ultrasound assessment of radial artery caliber and anatomical variants	21G or 18G needle, Ultrasound guidance, 11 cm 5F hydrophilic sheath (Prelude Ease, Merit Medical, South Jordan, UT, USA) placed over the guidance wire, Seldinger technique	-	Hemostasis technique with a Statseal hemostatic pad (Biolife, Sarasota, FL, USA) and a Safeguard radial hemostasis balloon (Merit Medical, South Jordan, UT, USA)
Gjoreski, 2019	Barbeau test Radial arteries diameter assessment by ultrasonography (>2.5 mm)	Micropuncture set for transradial access (5F Slender Glidesheath, Terumo, Japan)	110 cm MP or 125 angled catheters for cannulation, 150 cm long 2.8F microcatheter (Program, Terumo, Japan) in combination with GT microwire was chosen for uterine artery super selective catheterization,	TR Band (Terumo Interventional Systems)
Nakhaei, 2019	Barbeau test or modified Allen test	5F hydrophilic sheath (Prelude Ease, Merit Medical, South Jordan, UT, USA), Seldinger technique	5F angled catheter (Berenstein Performa catheter, 125 cm, Merit Medical, or Vertebral catheter, 125 cm, Cook Medical, Bloomington, Indiana) and J-tip Glidewire (Terumo, Somerset, NJ, USA). Maestro 150-cm microcatheter (Merit Medical) and microwire (Transcend 160 cm, Stryker Neurovascular, Fremont, CA, USA; or Fathom, 180 cm, Boston Scientific, Marlborough, MA, USA) were used to cannulate the horizontal component of the uterine artery	TR Band (Terumo Interventional Systems)

TRA—Transradial approach.

**Table 3 medicina-57-00083-t003:** Summary of TRA cases and technical success rate in patients with uterine fibroids.

Author, Year	Technical Success of TRA (%)	Number of UAEs Cases via TRA
Yamashita, 2007	95% of total 380 cases	7
Resnick, 2014	100	29
Salmeron, 2015	97	62
Posham, 2016	98.20	116
Thakor, 2016	99.50	88
Biederman, 2016	100	4
Pham, 2016	100	60
Mortensen, 2018	100	27
Gjoreski, 2019	100	11
Nakhaei, 2019	98.90	90
Total	-	494

TRA—Transradial approach; UAE—Uterine artery embolization.

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
