# Peer review of "Transradial Embolization, an Underused Type of Uterine Artery Embolization Approach: A Systematic Review"

_medicina, 2021, doi:10.3390/medicina57020083_

Round 1

Reviewer 1 Report

The authors adequately addressed the comments from reviewers.

Author Response

Dear Reviewer 1,

Thank you for the time and effort that you have dedicated to providing your valuable feedback on our manuscript! 

Sincerely, 

Loredana Himiniuc

MD, PhD student

Reviewer 2 Report

if they used abbreviations in tables, they should be explained below the tables.

Author Response

Dear Reviewer 2,

Thank you for your valuable remark on our paper. We agree with this and we have incorporated your suggestion throughout the manuscript. He have highlighted the changes within the manuscript by using red color. Thank you for your time and consideration!

Sincerely,
Loredana Himiniuc
MD, PhD student

This manuscript is a resubmission of an earlier submission. The following is a list of the peer review reports and author responses from that submission.

Round 1

Reviewer 1 Report

Interesting review on the radial access to perform uterine embolization.

The manuscript is well written and the review well conducted.

Please avoid the use of first person ("we", "our").

Intro: Needs to be thoroughly edited. second paragraph may be removed.Third paragraph: please add references. Fourth, please down the statements on complications related to femoral access: this is true but other access have also some disadvantages. The rationale of the review is to demonstrate the value of radial access, not the disadvantages of femoral route.

Material and methods: ok

Results: a paragraph on material used would be interesting (size of catheter sheeth, length, side, compression system...)

comparison with the transbrachial, transulnar or transfemoral approaces were not performed.

Discussion ok. But

Tables: need more details on technique for instance.

Reviewer 2 Report

Transradial embolization, an underused type ofuterine artery embolization approach: a systematicreview article

--> Transradial embolization, an underused type ofuterine artery embolization approach: a systematic review 

grey literature--> they should define what it is.

In table 1, the word in each cell should be start as capital letter and the world should not be cut.

Table 2, Author`s name--> Author, year